# The Role of a Polymer-Based E-Nose in the Detection of Head and Neck Cancer from Exhaled Breath

**DOI:** 10.3390/s22176485

**Published:** 2022-08-29

**Authors:** Roberta Anzivino, Pasqua Irene Sciancalepore, Silvano Dragonieri, Vitaliano Nicola Quaranta, Paolo Petrone, Domenico Petrone, Nicola Quaranta, Giovanna Elisiana Carpagnano

**Affiliations:** 1Otolaryngology Unit, Di Venere Hospital, 70124 Bari, Italy; 2Centre of Phoniatry and Rehabilitation of Communication Disorders-ASL Lecce, 73100 Lecce, Italy; 3Respiratory Diseases Unit, Department SMBNOS, University of Bari, 70121 Bari, Italy; 4General Directorate, ASL Bari, 70132 Bari, Italy; 5Otolaryngology Unit, Department SMBNOS, University of Bari, 70121 Bari, Italy

**Keywords:** electronic nose, e-nose, volatile organic compounds, exhaled breath, head and neck cancer

## Abstract

The aim of our study was to assess whether a polymer-based e-nose can distinguish head and neck cancer subjects from healthy controls, as well as from patients with allergic rhinitis. A total number of 45 subjects participated in this study. The first group was composed of 15 patients with histology confirmed diagnosis of head and neck cancer. The second group was made up of 15 patients with diagnoses of allergic rhinitis. The control group consisted of 15 subjects with a negative history of upper airways and/or chest symptoms. Exhaled breath was collected from all participants and sampled by a polymer-based e-nose (Cyranose 320, Sensigent, Pasadena, CA, USA). In the Principal Component Analysis plot, patients with head and neck cancer clustered distinctly from the controls as well as from patients with allergic rhinitis. Using canonical discriminant analysis, the three groups were discriminated, with a cross validated accuracy% of 75.1, *p* < 0.01. The area under the curve of the receiver operating characteristic curve for the discrimination between head and neck cancer patients and the other groups was 0.87. To conclude, e-nose technology has the potential for application in the diagnosis of head and neck cancer, being an easy, quick, non-invasive and cost-effective tool.

## 1. Introduction

Throughout the world, the term head and neck cancer (HNC) refers to tumors of the upper respiratory and digestive tract (tongue, oral cavity, oropharynx, nasopharynx, hypopharynx and larynx). These organs share the same squamous cell tissue and risk factors such as smoke and alcohol, posing a major public health problem [1]. Due to the disease heterogeneity and to the non-specificity of the symptoms, the diagnosis of HNC can be challenging, requiring tests that are often invasive, expensive and not broadly applicable [1]. Non-invasive and cost-effective measures of upper respiratory and digestive tract pathology would be ideal for the diagnostic investigation in any patients.

It is well-known that human exhaled breath includes a combination of around 3000 volatile organic compounds (VOCs) [2]. These VOCs initiate from both systemic and in-lung metabolic processes, as well as during disorders in the airways or anywhere else in the body. Only a small portion of VOCs have been identified as disease-related, including altered patterns of alkanes, methylalkanes, aldehydes, benzenes and alcohols [3]. Gas chromatography coupled to mass spectrometry (GC-MS) is the gold-standard method for measuring VOCs in exhaled breath, though it has several limitations, such as high costs, the need for well-trained personnel, the lack of standardized collecting methods and problems in detecting VOCs in concentrations lower than nanomolar [4,5,6,7]. The recently introduced electronic nose (e-nose) systems allow the real-time detection of VOCs by multiple nano-sensor arrays combined with potent learning algorithms. This high-throughput diagnostic methodology (also known as omics) leads to the appraisal of the biomarker spectrum, resulting in a specific “breathprint” of the disease [8]. There are several technologies for e-nose sensors, ranging from metal oxides, acoustic waves, colorimetric or optical sensors, gold nanoparticles and conducting polymers [9]. Most of these e-noses have been built not for medical purposes, but for a very widespread range of applications, including the food and military industries, which might be extended to medical applications.

Very recently, several authors have shown the potential of exhaled breath molecular profiling in the medical field, including neoplasms such as Lung cancer and Malignant Pleural Mesothelioma along with breast cancer, ovarian cancer, and colon cancer [10,11].

Regarding HNC, proof of concept studies have demonstrated that a metal oxide- based e-nose can distinguish head and neck squamous cell carcinoma subjects from healthy controls [12,13]. We postulated that this discrimination can also be accomplished in a heterogeneous group of patients with HNC using a polymer-based e-nose. This pilot study sought to explore the above-mentioned hypothesis.

## 2. Materials and Methods

### 2.1. Patients

A total number of 45 subjects participated in this study. Patients were recruited from the otolaryngology outpatient clinic of the Di Venere Hospital, Bari, Italy, whilst controls were recruited amongst personal contacts of the authors. The recruiting period was June 2020–July 2022.

The first group was composed of 15 patients with HNC. Diagnosis was established by histology on pre-operatory biopsies without prior treatment by radiotherapy and/or chemotherapy. All individuals received a computed tomography (CT) scan of neck and chest and a rhino-laryngoscopy.

The second group consisted of 15 individuals with intermittent to persistent symptoms of allergic rhinitis without episodic or chronic chest symptoms and positive skin prick tests or positive RAST tests.

The control group comprised 15 subjects without a clinical history of upper and lower respiratory tract diseases.

Participants who experienced upper or lower respiratory tract infections in the 28 days prior to the measurements and patients with systemic diseases, including diabetes or cardiac failure, or with a precedent diagnosis of malignancy were excluded from this study. No food or drinks (except water) or physical exercise were allowed in the 3 h preceding the breath collection. Smoking was not permitted during the 24 h before testing.

Oral informed consent was obtained from all patients. The study protocol was approved by the local medical ethics committee in accordance with the Declaration of Helsinki.

### 2.2. Study Design

We performed a cross-sectional case-control study with two visits within a 10-day period. Day one was a screening day to check all the inclusion and exclusion criteria. On day two, exhaled breath was obtained and sampled by the e-nose within 5 min of the collection.

### 2.3. Breath Collection and Sampling

After 5 min of tidal breathing through a mouthpiece connected to a 2-way non-rebreathing valve (1410 series, Hans Rudolph, Kansas City, KS, USA) with inspiratory VOC-filter (A2 type, North Safety, Middelburg, The Netherlands), the subjects performed a single deep inspiration followed by an expiratory vital capacity manoeuvre into a 5-litre Tedlar bag (SKC, Mantua, Italy). Within 10 min, the e-nose (Cyranose 320, Sensigent, Irwindale, CA, USA) was connected to the bag followed by a 1 min sampling of the expired air. VOC-filtered room air was used as the baseline (Figure 1).

### 2.4. Electronic Nose

A Cyranose 320 (Sensigent, Pasadena, CA, USA) was used for our study.

This is a commercially available handheld portable chemical vapor analyzer, containing a nano-composite array with 32 polymer sensors. When exposed to a gas mixture the sensors swell, thereby increasing their electrical resistance (ΔR/R). The combination of the 32 sensors ΔR/R results in a unique pattern, named “breathprint”, which can be stored and compared with others (Figure 2) [14].

### 2.5. Data Analysis

The breathprints of the exhaled breath samples were analyzed by SPSS software, version 21.0. The sample size was estimated to limit the standard error to 10%. Assuming 80% accuracy, the current sample size per subgroup was adequate.

Principal component analysis (PCA) and subsequent linear canonical discriminant analysis (CDA) were conducted, providing the cross validated accuracy percentage (CVA%), which estimates how accurately a predictive model will perform in practice. Additionally, a receiver operating characteristic curve (ROC-curve) was built using predicted probabilities to determine the area under the curve (AUC). A *p*-value of < 0.05 was considered significant.

## 3. Results

Subject characteristics of the three groups are shown in Table 1. Slight imbalances in terms of sex, age and pack-years among the groups were reported.

The clinical characteristics of patients with HNC are shown in Table 2. In 8 out of 15 patients, the tumor was less than 4 cm in its greatest dimension, whereas 7 patients had a locally advanced disease, and 5 individuals had lymph node involvement.

Although PC2 explained 17.5% of the total variance and PC3 7.5%, ANOVA on PC3 was more significant than on PC2 (*p* = 0.000 vs. *p* = 0.001, respectively, see Table 3). Therefore, we chose to use PC1 and PC3 for further discriminant analysis.

In the PCA plot, patients with HNC clustered distinctly from the controls, as well as individuals with allergic rhinitis (Figure 3). Using CDA, the two groups were discriminated, with a CVA% of 75.3, *p* < 0.01. The AUC of the ROC curve for the discrimination between HNC patients and the other groups was 0.87 (Figure 4).

By means of the ROC curve, we used the Youden Index method to obtain the best cut-off of the model (0.4831760) in discriminating patients with HNC. Consequently, we calculated a sensitivity of 93.3%, a specificity of 86.6%, a positive predictive value (PPV) of 77.8% and a negative predictive value (NPV) of 96.2%

When performing group-to-group comparisons, the following CVA% were obtained:

HNC vs. healthy controls: 94.3%; HNC vs. allergic rhinitis: 74.3%; healthy controls vs. allergic rhinitis: 75.2%.

## 4. Discussion

In the current pilot study, we aimed to assess the feasibility and potential of a polymer-based e-nose for diagnosing HNC in a clinical setting.

Based on our results, we conclude that the e-nose can discriminate exhaled breath of patients with HNC from controls, as well as from individuals with allergic rhinitis. To the best of our knowledge, this is the first study where exhaled breath samples collected from well-characterized patients with and without heterogeneous forms of HNC were analyzed by a polymer-based e-nose. Our data extend the previous studies applying VOC-profiling analysis in HNC. Leunis et al. [12] studied the diagnostic accuracy of VOC-pattern analysis in exhaled breath by means of a metal oxide-based e-nose in patients with head and neck squamous cell carcinoma, showing a significant difference in VOC resistance patterns between patients with neoplasm and the control group, with a sensitivity of 90% and a specificity of 80% [12]. Moreover, using a similar technology to that above, van der Goor et al. [13] assessed the diagnostic performance of an e-nose in detecting locoregional recurrent head and neck squamous cell carcinoma after curative treatment, evidencing a diagnostic accuracy of 83% in discriminating follow-up patients with locoregional recurrent or second (or third) primary tumor from those without evidence of disease [13]. Furthermore, the same group applied e-nose technology to distinguish healthy patients from patients with primary head and neck squamous cell carcinoma and bladder cancer, colon carcinoma and lung carcinoma [15,16]. The present findings confirm the potential of e-nose technology to become a diagnostic tool for HNC.

The strengths of our study are the meticulous selection of patients and controls by worldwide accepted guidelines [17]. This approach is crucial when probing discriminative potential of a new test. In addition, we used previously validated techniques for breath collection and sampling [18].

Differences in sex and age among our groups may have altered our findings. However, our group has previously demonstrated that age and gender do not seem to affect the overall profile of exhaled VOCs measured by an e-nose [19].

Lastly, tobacco smoking may play a pivotal role in altering the VOC profile in human breath [20]. To minimize the impact of smoke on exhaled VOCs between our HNC group and the control group, all included subjects had a positive history of smoking and all participants abstained from smoking during the 24 h preceding the test. However, also due to imbalances in pack-year among the groups, we cannot completely rule out the possible influence of smoke on our results. The main limitation of our study is due to the small population. Nonetheless, 15 individuals per group might be sufficient for obtaining a discrimination between HNC patients and controls. In addition, an implicit limitation of our study is the lack of data about the identification of specific VOCs. However, the need to identify discriminant VOCs might not be essential when evaluating the usefulness of e-noses in clinical practice. Undoubtedly, the identification and quantification of exhaled VOCs would be important for gaining knowledge of the biologic nature of HNC and for contributing to the development of e-noses with more specific sensors.

Unsurprisingly, when performing group-to-group comparisons, the distinction between exhaled VOC spectrum in HNC patients and subjects with allergic rhinitis was less sharp than in HNC vs. healthy controls. This could be explained by the underlying inflammation of the nasal mucosa that might generate VOCs resulting from oxidative stress processes [21]. Similar VOCs can also be produced during neoplastic processes, deriving from increased production of reactive oxygen species and enhanced alkane metabolism by cytochrome P450 [22].

To conclude, an e-nose is a handheld, quick, non-invasive and cost-effective tool that has shown its potential in the diagnostic process of HNC. Strictly following the current guidelines in the validation of new diagnostic tests [23], our findings warrant further studies aiming at the comparison of larger populations with newly presented patients with HNC. Moreover, integration with other types of e-nose and VOCs identification by gas chromatography and mass spectrometry is mandatory.

## Figures and Tables

**Figure 1 sensors-22-06485-f001:**
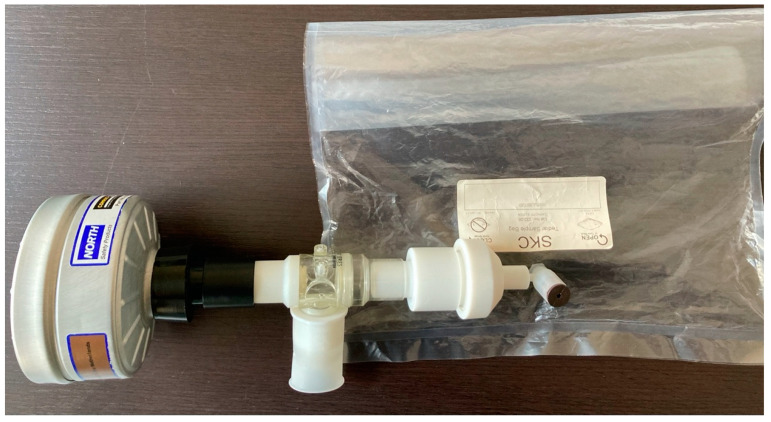
Setup for breath collection. From left to right: A2-environmental VOCs filter; 2-way non rebreathing valve; expiratory port; tedlar bag.

**Figure 2 sensors-22-06485-f002:**
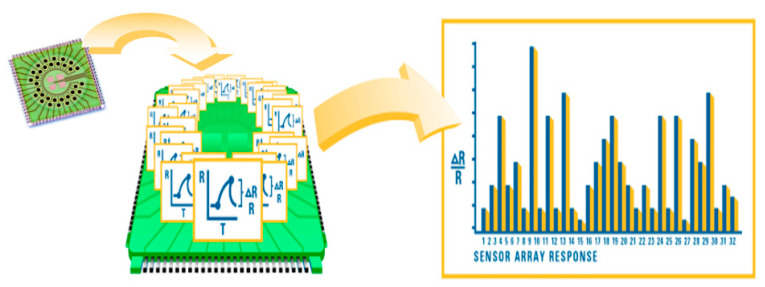
Working principle of Cyranose 320: it is based on a nano-composite array of 32 organic polymer sensors. If exposed to VOC combinations, the polymers swell, thereby modifying their electrical resistance. Raw data are registered as the increase in resistance of any single sensors and the combination of all signals results in a “breathprint”.

**Figure 3 sensors-22-06485-f003:**
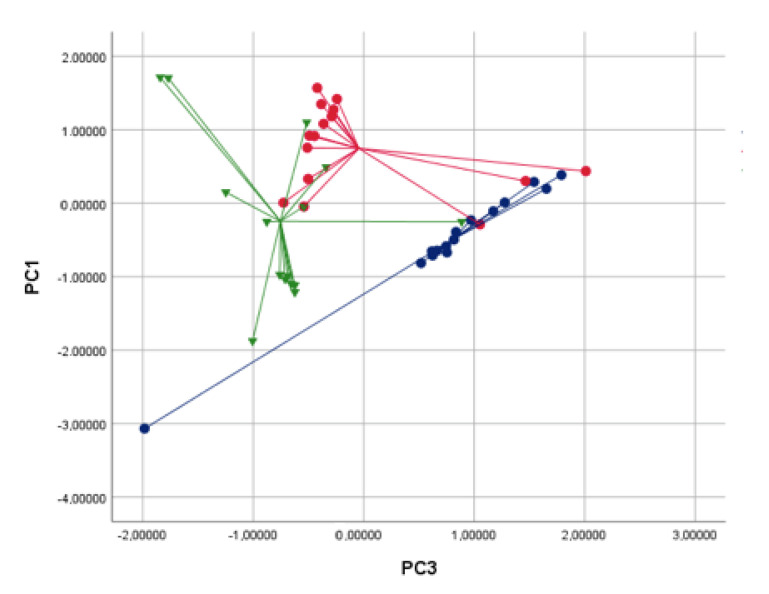
Two-dimensional PCA with 2 composite factors (x axis = Principal Component 3; y axis = Principal Component 1) showing the discrimination between patients with HNC (blue circles), allergic rhinitis (red squares) and healthy controls (green triangles). Cross-validated accuracy was 75.3% (*p* < 0.01).

**Figure 4 sensors-22-06485-f004:**
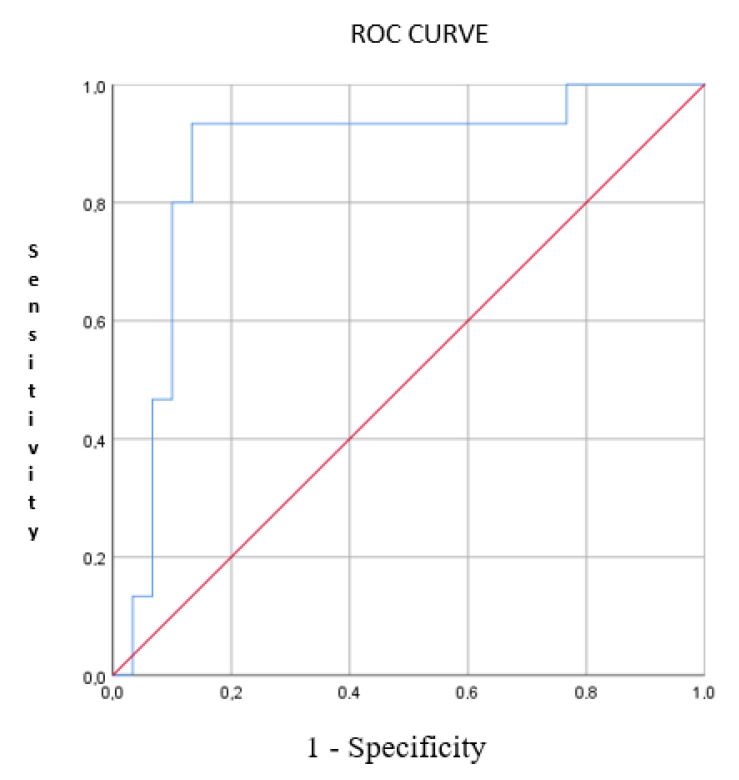
ROC-curve with line of identity of the breathprint discriminant function (representing PC1 and PC3), predictive for the discrimination of HNC from allergic rhinitis and healthy controls. AUC was 0.871 (95% CI: 0.749–0.994). Blue line = ROC-curve; Red line = random classifier.

**Table 1 sensors-22-06485-t001:** Clinical characteristics of the study population. Values are intended as mean ± SD.

	HNC	Allergic Rhinitis	Controls
Subjects (n.)	15	15	15
M/F (n.)	10\5	7\8	8\7
Age (y.)	61 ± 12	48 ± 13	51 ± 11
Pack-Years	35 ± 16	24 ± 12	24 ± 13

**Table 2 sensors-22-06485-t002:** Histology and TNM stage for the group of patients with HNC. T = tumor size; N = lymph nodes involvement; M = metastasis yes/no.

Patient	Tumor Type	TNM Stage
1	Larynx squamous cell carcinoma	T4a N0 M0
2	Larynx squamous cell carcinoma	T3 N0 M0
3	Tonsillary carcinoma	T2 N1 M0
4	Larynx lymphoepithelial carcinoma	T3 N1 M0
5	Tonsillary carcinoma	T2 N0 M0
6	Tongue carcinoma	T2 N0 M0
7	Larynx squamous cell carcinoma	T3 N1 M0
8	Tonsillary carcinoma	T2 N0 M0
9	Tonsillary carcinoma	T1 N0 M0
10	Epiglottis squamous cell carcinoma	T3 N0 M0
11	Larynx squamous cell carcinoma	T2 N0 M0
12	Tongue carcinoma	T3 N3b M0
13	Tongue carcinoma	T3 N0 M0
14	Larynx lymphoepithelial carcinoma	T2 N1 M0
15	Sinonasal squamous cell carcinoma	T2 N0 M0

**Table 3 sensors-22-06485-t003:** ANOVA of the four principal components (PC) among the three groups.

Principal Component	HNC	Allergic Rhinitis	Controls	*p*
PC1 (72% of total variance)	−0.5 ± 0.8	0.7 ± 0.6	−0.2 ± −1.1	0.001
PC2 (17% of total variance)	0.2 ± 0.3	−0.7 ± 0.6	0.5 ± 1.3	0.001
PC3 (7.5% of total variance)	0.8 ± 0.9	−0.4 ± 0.8	−0.7 ± 0.6	0.000
PC4 (1% of total variance)	0.4 ± 0.8	0.1 ± 0.8	−0.7 ± 1.2	0.721

## Data Availability

Not applicable.

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
