# Peer review of "The Role of a Polymer-Based E-Nose in the Detection of Head and Neck Cancer from Exhaled Breath"

_sensors, 2022, doi:10.3390/s22176485_

Round 1

Reviewer 1 Report

In my opinion, the 32 polymer scheme should be presented in more details, as should the difference in results for all potentially discriminating polymers, if available by the product specifications. This issue is addressed in the discussion, but the quality of paper would be improved with this. 

The study group is quite small for strong conclusions to be made, but it shows a potential for futher studies. 

Author Response

R- Many thanks for your constructive comments. We have added a new figure which better explains the way of functioning of the 32 polymer scheme (please see new figure 2). Unfortunately, we do not have the results for all potentially discriminating polymers because they are not available by the product specifications.

In addition, we have added a new control group of patients with allergic rhinitis so we have increased our sample size

Reviewer 2 Report

The communication is interesting even if it needs some more technical  details that could better tailoring it to the aims of the journal .

Explain the choice of Cyranose 320 and its pros/cons with respect other technologies

Figure 1: explain what are the variables used for this figure

Figure 2: explain what is the meaning of continuous/dashed lines

Table 1: row 2  check the backslash; row 3-4 the HCN and the control populations seems to belong to too different classes: they canno be compared either by age or by pack-years . The results (PCA) may  simply detect younger and no/low smokers instead of hill/safe individuals.

Table 2: explain the meaning of the T*, N*, M* symbols

Author Response

The communication is interesting even if it needs some more technical  details that could better tailoring it to the aims of the journal .

Explain the choice of Cyranose 320 and its pros/cons with respect other technologies.

R- Thanks for your comment. We have added in introduction section a review of all e-nose available technologies with new references. To be honest, the choice of Cyranose was due to the fact that it is the one available in our clinic. However, we have previously shown that cyranose is able to distinguish the exhaled breath of patients with several respiratory diseases (you can find previous literature at this link: https://pubmed.ncbi.nlm.nih.gov/?term=dragonieri+electronic+nose&sort=date). Therefore, we rely on our device, although we cannot really state whether is more or less effective than other e-noses. Future studies should be focused on the comparison of different e-noses diagnostic powers in (for example) lung cancer or other diseases.  

Figure 1: explain what are the variables used for this figure

Figure 2: explain what is the meaning of continuous/dashed lines

Table 1: row 2  check the backslash; row 3-4 the HCN and the control populations seems to belong to too different classes: they canno be compared either by age or by pack-years . The results (PCA) may  simply detect younger and no/low smokers instead of hill/safe individuals.

R- Thanks for your comments. According to another reviewer’s comments we have now modified our population by adding a new control group of patients with allergic rhinitis and made new analysis. Please see methods and results sections.

Table 2: explain the meaning of the T*, N*, M* symbols

R- Done it. Thank you.

Reviewer 3 Report

In the manuscript entitled “A polymer-based e-nose discriminates the exhaled breath of patients with head and neck cancer from controls”, R. Anzivino et al. have reported a study regarding the use of polymer-based e-noise to detect the exhaled breath of patients affected of head and neck squamous cell carcinoma, as a non-invasive and cost-effective technology.

First of all, the title is written in a wrong format, a sentence! Please, re-write it in the correct form.

In the abstract the authors should briefly indicate the aim and the objectives of the reported study; therefore, reduce it, eliminate the first paragraph and start from “The aim …”.

Do not use acronyms in the abstract.

Add more references in the introduction.

In the introduction the authors should give a brief information of VOCs, reporting some examples; moreover, they should indicate which are the common technologies used to monitor and/or detect VOCs from metabolic processes.

In the introduction, the authors should briefly discuss about e-nose systems, indicating the basis of this technology and reporting some examples of applications, adding correct references.

In the introduction, the authors should report common used materials in e-nose, adding the advantages and disadvantages.

In the introduction, the authors should describe the common polymers used for e-nose, reporting their application, and their advantages and disadvantages.

Please, give more information about the breath collection and sampling, on the used VOCs filter. Add a scheme of the process.

In the experimental section, the authors should give more information on the used e-nose, adding information on the nanocomposite array, on the 32 used polymers. Add a scheme of the array, explaining how it works, etc.

In the plot in Figure 1, the authors should express Fact. 1 and Fact. 2.

The authors should explain the KOC-curve.

The authors should define sensitivity and selectivity, towards what? Discuss.

The author should describe and discuss all reported plots and tables.

The authors indicate VOCs in general, but I suggest to specify which type of VOCs are detected, which are their concentration detectable ranges, and other experimental condition.

Moreover, which is the sensor measurements reproducibility, etc.

The English style is poor.

I reject the manuscript in the present form.

Author Response

R- Many thanks for your comments which helped us to improve the quality of our manuscript. Please take a look at the revised version, where we tried to address to all your comments, by modifying the title and the abstract, improving introduction section, adding new figures and adding a new control group made by patients with allergic rhinitis and with new data analysis. Finally, we had a language revision.

Reviewer 4 Report

Sensors (ISSN 1424-8220)

Manuscript ID: sensors-1787262

Type: Communication

Title: A polymer-based e-nose discriminates the exhaled breath of patients with head and neck cancer from controls.

Anzivino et al.

This communication describes a pilot study that investigated the validity of a commercial e-nose (Cyranose 320) for HNC diagnosis. Even if acknowledging that it is not an original research manuscript but a communication manuscript, and moreover, it is a pilot study, not a full-scale research and development, it still leaves much to be desired as a publishable manuscript in Sensors Journal. First, and the foremost, the test conditions are insufficient. There are two groups representing the HNC patients and healthy patients, whereas another positive control group should have been included. For example, a group of patients who has different type of cancer. Interestingly, and ironically, this issue was actually recognized and mentioned by the authors in the discussion. Another significant issue, which was also recognized and mentioned by the authors, is the factor of smoking that may have influence the detection result. The authors maintained that these issues can be probated because the manuscript is a communication and it was for proof-of-concept. Be that as it may, these two issues must be ironed out in order for this communication to be published in Sensors as a communication manuscript. A communication manuscript does not mean to have a reprieve of required test conditions.

The manuscript also suffers from lack of clear explanation on the data. Again, knowing that it is meant to be a communication which has the length and the number of words limitations, the variables in the plots such as Figures 1 and 2 should be briefly provided. What are Factors 1 and 2 composed of? How were the sensitivity and 1-specificity estimated and plotted? 

The reviewer agrees with the authors on their argument that the patients were well-characterized. However, as the authors admitted, the number of samples is far lower than the statistically valid value. Roughly it can be easily calculated that at least about 40 populations are needed to have 80% accuracy and 10% standard error with a p-value less than 0.05. Fifteen patients are not enough to come up with a conclusive result. At one point, the manuscript says the number of samples is adequate (p. 3), and in another point (p. 7), it says the number of samples is a main limitation. The authors themselves were conflicted with the number of samples.

There are several typos in this short communication that derogate the credibility of the manuscript.

Author Response

R- Many thanks for your comments which helped us to improve the quality of the manuscript. We have now added a new control group of patients with allergic rhinitis (raising up the sample size to 45 patients) and made new analysis which is now more detailed. Moreover, we have done a language revision and corrected several errors.

Reviewer 5 Report

The authors present a study on the testing of a commercial polymer-based e-nose system with healthy persons and patients with HNC. The study seems well conducted and conclusive. Before publication I would advise a thorough spell/grammar check, since there are still some mistakes in the text.

Based on the fact that it is a preliminary study and rather short, the type of the paper might be changed in "communication" instead of "full article".

Author Response

R- Many thanks for your comments. We have done a language revision and corrected several errors and changed to communication.

Round 2

Reviewer 2 Report

Explain why you used PCA3 instead of PCA2. Discrimination does not seem too effective, mainly between red and green data. The blue data are, otherwise well identified.

Concerning Table 1, due to the great uncertainty on data (> 20%) the correct way to report them is the following:

Subject : ok

M/F    

10/5

7/8

8/7

Age (yr)

61±12

48±12

51±11

Pack-Years

35±16

24±12

24±13

Modify accordingly also Table 3

Author Response

Explain why you used PCA3 instead of PCA2. Discrimination does not seem too effective, mainly between red and green data. The blue data are, otherwise well identified.

R- Thanks for your question. Although PC2 explained 17.5% of the total variance and PC3 only 7.5%, ANOVA on PC3 was more significant than on PC2 (p=0.000 vs p=0.001, respectively). Therefore, we decided to use PC3. This is something that may occur when using Principal component analysis and subsequent discriminant analysis (less variance/more discrimination). Moreover, when performing discriminant analysis with PC1 e PC2, results were poorer in terms of discrimination compared to that using PC1 and PC3. (Cross Validated accuracy 62.1% vs 75.3% and AUC 0.75 vs 0.87, respectively).

Concerning discrimination about the groups, you are right. When performing group to group comparison, we have the following results:

  • HNC (blue) vs healthy controls (green): 94.3%
  • HNC (blue) vs allergic rhinitis (red): 74.3%
  • Healthy controls (green) vs allergic rhinitis: 75.2%

We are not surprised about the above findings, which are likely due to the underlying inflammation of the nasal mucosa that might generate VOCs resulting from oxidative stress processes. Similar VOCs can also be produced during neoplastic processes, deriving from increased production of reactive oxygen species and enhanced alkane metabolism by cytochrome P450. We have now addressed the above by expanding results and discussion section and by adding new references [21 and 22].

Concerning Table 1, due to the great uncertainty on data (> 20%) the correct way to report them is the following:

Subject : ok

M/F    

10/5

7/8

8/7

Age (yr)

61±12

48±12

51±11

Pack-Years

35±16

24±12

24±13

Modify accordingly also Table 3

R- Thanks for your advice. We have modified tables accordingly.

Reviewer 3 Report

In the revised manuscript, the authors have not answered punctually to all reviewer comments; therefore, the authors should attach the punctual responses, indicating the variations in the text.

In my opinion, a question as title should not be use; please, modify

In the revised version, a lot of lacks are still present:

In the introduction, the authors should briefly discuss about e-nose systems, indicating the basis of this technology and reporting some examples of applications, adding correct references.

In the introduction, the authors should report commonly used materials in e-nose, adding the advantages and disadvantages.

In the introduction, the authors should describe the common polymers used for e-nose, reporting their application, and their advantages and disadvantages.

Please, give more information about the breath collection and sampling, on the used VOCs filter. Add a scheme of the process.

In the experimental section, the authors should give more information on the used e-nose, adding information on the nanocomposite array, on the 32 used polymers. Add a scheme of the array, explaining how it works, etc.

The authors should explain the ROC-curve.

The authors should define sensitivity and selectivity, towards what? Discuss.

The author should describe and discuss all reported plots and tables.

The authors indicate VOCs in general, but I suggest specifying which type of VOCs are detected, which are their concentration detectable ranges, and other experimental condition.

Moreover, which is the sensor measurements reproducibility, etc.

Since all these lacks still present, I can not accept this revised version.

Author Response

In the revised manuscript, the authors have not answered punctually to all reviewer comments; therefore, the authors should attach the punctual responses, indicating the variations in the text.

R- We apologize with the reviewer for not being accurate in our replies. We will now try to provide point to point responses.

In my opinion, a question as title should not be use; please, modify

R- Thanks for your suggestion. We have now modified the title as follows: The role of a polymer-based e-nose in the detection of head and neck cancer from exhaled breath.

In the revised version, a lot of lacks are still present:

In the introduction, the authors should briefly discuss about e-nose systems, indicating the basis of this technology and reporting some examples of applications, adding correct references.

R- Thanks for your suggestion. We have expanded introduction section accordingly (please see highlighted lines 52-56 and new reference 9).

In the introduction, the authors should report commonly used materials in e-nose, adding the advantages and disadvantages.

R- Thanks for your suggestion. We have added in lines 52-56 commonly used materials in e-nose. However, due to the nature of the paper (a communication) and being us all medical doctors with a little technical knowledge about e-nose technology, we do not have the skills to provide advantages and disadvantages. Furthermore, we humbly believe that adding too many technical details would drift the focus from a “more medical” paper to a “more technical” one. We hope you understand our point of view.  

Please, give more information about the breath collection and sampling, on the used VOCs filter. Add a scheme of the process.

R- Thanks for your suggestion. We have now added new figure 1, which explains how we collected exhaled breath and analyzed it.

In the experimental section, the authors should give more information on the used e-nose, adding information on the nanocomposite array, on the 32 used polymers. Add a scheme of the array, explaining how it works, etc.

R- Many thanks for your very constructive comment. We have now added new  figure 1, which explains how the sensor array work .

The authors should explain the ROC-curve.

R- Thanks for your advice. We have now better explained ROC-curve (please see figure 4).

The authors should define sensitivity and selectivity, towards what? Discuss.

R- Thanks for your suggestion. By means of the roc curve we obtained with the Youden Index method the best cutoff of the model (0.4831760) in discriminating patients with HNC. Thus, we calculated a sensitivity of 93.3%, a specificity of 86.6%, a positive predictive value (PPV) of 77.8% and a negative predictive value (NPV) of 96.2%. We have added the above in result section.

The author should describe and discuss all reported plots and tables.

R- Thanks for your suggestion. We have now implemented plots and tables and added discussion.

The authors indicate VOCs in general, but I suggest specifying which type of VOCs are detected, which are their concentration detectable ranges, and other experimental condition. Moreover, which is the sensor measurements reproducibility, etc.

R- This is a very interesting suggestion. The electronic nose that we used detects VOC mixtures with a polymer nanocomposite sensor array. Each sensor signal represents partly different fractions of the complete VOC mixture based on, for example, molecular mass, shape, dipole moment, and hydrogen binding capacity. Therefore, unlike GC-MS, an electronic nose principally does not chemically identify and separate VOCs. Whether it contributes to the discrimination between patient groups depends on any differences in this signal between the groups and its subsequent selection by the PCA. Therefore, pattern recognition of breathprints by electronic noses is purely based on a statistical approach, providing empirical evidence. Such procedure is hypothesis-free and potentially powerful. It resembles other high-throughput methods that are based on the analysis of molecular profiles of complex biological samples by a single measurement (omics techniques). Interestingly, the principle of electronic noses exactly mirrors biological olfaction in mammals, in which multisensitive olfactory receptor cells appear to be coupled to pattern recognition systems in the brain, leading to unique odors. The downside is that we cannot identify single VOCs as well as concentration ranges and other experimental conditions. For that we would need further studies by coupling GC-MS. Hence, our current observations by using an electronic nose system can be considered complementary to GC-MS analysis and warrant further studies by GC-MS to identify the critical VOCs.

Reviewer 4 Report

The communication manuscript has been significantly improved in the current revision by adding a group of 15 allergic patients.

Please consider the following points in further revision:

p.3, line 137: 4cm ---> 4 cm

p.5, Figure3: “REGR” on the x and y axis is supposed to mean “Regression” as in PCA, but it should be fully expanded on the figure or in the text.

p.6 Figure 4: Legends or brief explanations on the blue line and red line should be added.

Author Response

The communication manuscript has been significantly improved in the current revision by adding a group of 15 allergic patients.

Please consider the following points in further revision:

p.3, line 137: 4cm ---> 4 cm

p.5, Figure3: “REGR” on the x and y axis is supposed to mean “Regression” as in PCA, but it should be fully expanded on the figure or in the text.

p.6 Figure 4: Legends or brief explanations on the blue line and red line should be added.

R- Many thanks for your comments. We have addressed in the manuscript all your suggestions.